# Simple and Cost-Effective Electrochemical Method for Norepinephrine Determination Based on Carbon Dots and Tyrosinase

**DOI:** 10.3390/s20164567

**Published:** 2020-08-14

**Authors:** Sylwia Baluta, Anna Lesiak, Joanna Cabaj

**Affiliations:** 1Faculty of Chemistry, Wrocław University of Science and Technology, Wybrzeże Wyspiańskiego 27, 50-370 Wrocław, Poland; sylwia.baluta@pwr.edu.pl (S.B.); anna.lesiak@pwr.edu.pl (A.L.); 2Faculty of Fundamental Problems of Technology, Wrocław University of Science and Technology, Wybrzeże Wyspiańskiego 27, 50-370 Wrocław, Poland

**Keywords:** biosensor, carbon dots, norepinephrine, tyrosinase, voltammetry

## Abstract

Although neurotransmitters are present in human serum at the nM level, any dysfunction of the catecholamines concentration may lead to numerous serious health problems. Due to this fact, rapid and sensitive catecholamines detection is extremely important in modern medicine. However, there is no device that would measure the concentration of these compounds in body fluids. The main goal of the present study is to design a simple as possible, cost-effective new biosensor-based system for the detection of neurotransmitters, using nontoxic reagents. The miniature Au-E biosensor was designed and constructed through the immobilization of tyrosinase on an electroactive layer of cysteamine and carbon nanoparticles covering the gold electrode. This sensing arrangement utilized the catalytic oxidation of norepinephrine (NE) to NE quinone, measured with voltammetric techniques: cyclic voltammetry and differential pulse voltammetry. The prepared bio-system exhibited good parameters: a broad linear range (1–200 μM), limit of detection equal to 196 nM, limit of quantification equal to 312 nM, and high selectivity and sensitivity. It is noteworthy that described method was successfully applied for NE determination in real samples.

## 1. Introduction

One of the primary goals of worldwide scientific endeavors is to improve quality of life. Achieving this is directly related to the rapid analysis of common disorders, quality control in the food industry and environment monitoring. Constant, fast and sensitive in situ monitoring is a priority in diagnostic control, and most of all, in medical diagnostics. The devices that meet these requirements are biosensors. According to Cammann, biosensors are analytical devices allowing for conversion of a biological signal to a measurable signal, like for instance amperometric response in the case of electrochemical sensors [1].

In electrochemical biosensors, differential pulse voltammetry (DPV) is frequently used. DPV applies a linear sweep voltammetry with a series of regular voltage pulses superimposed on the linear potential sweep [2]. Due to this, the current can be estimated instantly before every change of potential. In consequence, the influence of the charging current is minimized, reaching a better sensitivity [3]. DPV is frequently used in voltammetry-based techniques, not only due to its good sensitivity, but also because of resolving power.

Norepinehrine (NE) is a monoamine neurotransmitter engaged in a broad range of physiological actions. The main function of NE is to help the organism adapt to internal and external environmental changes. Noradrenergic neurons influence many effects in the human body, such as processing sensory information, regulating sleep, and mediating attentional functions [4,5,6]. Any changes in NE level may directly affect the mental state, neurodegenerative disorders or cardiovascular function [7]. Therefore, NE monitoring is crucial from a medical point of view. 

Many of the electrochemical biosensors for neurotransmitter detection, e.g., dopamine (DA), epinephrine (EP), and NE utilize oxidoreductases as a recognition element [8,9,10,11,12]. For instance, tyrosinase and laccase are multifunctional copper containing enzymes with two types of catalytic activity in the presence of oxygen: hydroxylation of monophenols to *o*-diphenols by cresolase activity, and oxidation of diphenols to *o*-quinones by catecholase activity [13]. The resulting quinones can be further reduced electrochemically on the electrode without any mediator, reforming the original *o*-diphenol. This reaction constitutes the basis of amperometric detection at negative potentials and quantification of phenolic compounds. By the same principle, catechol-like phenolic compounds (e.g., NE) can be detected by the amperometric method with an electrode modified with oxidoreductases [14]. There are also systems without any biologically active compound for NE detection, such as the most recent work presented by D. Ji et al., who demonstrated an electrochemical, smartphone-based system for NE detection [15]. Screen-printed graphene electrodes were modified, e.g., with GOx, and used as working electrodes. NE has been determined with square wave voltammetry (SWV) in the concentration range 1–30 µM. The detection limit obtained for such system was equal to 265 nM. The main advantage of the described system is the facility of miniaturization and the possibility of constant measurement, as a possible wearable device. It also presents the ability to work over a longer period of time (in comparison with biological systems).

Carbon dots (CDs) are an interesting platform for sensors because of their characteristic properties, such as water solubility, low toxicity, high emission intensity or chemical stability in time [16]. CDs’ structure consists of a carbon core and the surface passivation layer with functional groups (mainly hydroxyl-, carboxyl- or amine groups), which allow a conjugation with other molecules (like proteins) [17].

Data from the literature show the use of the CDs matrix to determine various substances, with the focus on their fluorescent properties, e.g., hydrazine in water [18]. The use of CDs in electrochemical sensors is slowly gaining popularity, among others, for the determination of ascorbic acid [19] or riboflavin [20].

In addition, an increase in interest in electrochemical methods based on nanoparticles for neurotransmitter determination has been observed [21,22,23,24,25]. Samdani et al. presented an electrochemical method for NE determination using a glassy carbon electrode (GCE), modified with nanorods based on FeMoO_4_. Synthesized nanorods were used as an active electrode material for the oxidation of NE by cyclic voltammetry (CV) and DPV techniques. The amperometric response of NE to the GCE/FeMoO_4_ nanorods showed a linear increase in the current between 5.0 × 10^−8^ M and 2.0 × 10^−4^ M with a detection limit of 3.7 × 10^−9^ M [26]. Another detection method for NE was investigated by Mohammadi et al. A glassy carbon electrode was modified with single-walled carbon nanotubes (SWCNTs) and immobilized tyrosinase. The DPV technique for NE determination was applied. The detection limit of the modified electrode towards NE was found at 0.1 μM, and the calibration curves were linear over the concentration range of 1.0–21 μM [27]. 

In this paper, we present a CDs-based electrochemical biosensor for NE determination, using a tyrosinase-dependent redox system. The described novel technique has many advantages, such as: the application of stable materials, synthesis of nanomaterials with nontoxic and inexpensive materials, simple and quick analysis, high sensitivity, good selectivity and measurements in a broad linear range. The designed biosensor is as simple as possible to reduce costs and to avoid using toxic species, which is essential for the medical or diagnostic industry, such as point-of-care (POC) testing. The gold electrode modified with cysteamine, CDs and tyrosinase (Tyr) presents excellent electrochemical behavior and provides a facile, selective and sensitive method, indicating that the biosensor is a good candidate for catecholamines detection.

## 2. Materials and Methods

### 2.1. Reagents and Materials 

Tyrosinase (from *Agaricus bisporus*, EC 1.14.18.1, ≥1000 U/mg) as well as norepinephrine hydrochloride (NE), epinephrine hydrochloride (EP), dopamine hydrochloride (DA), cysteamine (CA), glutaraldehyde (GA), uric acid (UA), ascorbic acid (AA), L-cysteine (CYS) and 4-*tert*-Butylcatechol (4*t*BC) were purchased from Sigma-Aldrich Co (Poznań, Poland). Citric acid, NaOH, NaH_2_PO_4_, KH_2_PO_4_, Tris, HCl, CH_3_COONa, CH_3_COOH, Na_2_HPO_4_, and K_2_HPO_4_ were purchased from POCH (Part of Avantor, Performance Materials, Gliwice, Poland). All chemicals were of analytical grade and were not further purified before use. All buffers (phosphate buffer, acetate buffer and Tris-HCl) were prepared according to commonly known, obligatory standards. In Section 2.2.2. the specific concentrations and pH values of prepared buffer solutions are listed. Synthetic urine CLEANU^®^ was produced by CleanU (Poznań, Poland), CU-25 mL, and it contained creatinine, uric acid, urea, mineral salts, dyes and water (ingredients reserved—patent).

### 2.2. Apparatus and Procedures 

#### 2.2.1. Synthesis and Characterization of CDs

The preparation process for CDs was carried out following a modified procedure of Sahu et al. [28]. Briefly, 10 mL of 100% orange juice was added to 20 mL of ethanol and the mixture was vigorously stirred for 60 min (at 75 °C). Afterwards, the mixture was cooled down to room temperature and 10 mL of dichloromethane was added. Subsequently, the mixture was centrifuged 3 times at 6000 rpm for 10 min to dispose of unreacted organic compounds. From the resulting biphasic solution, the aqueous layer was collected and 10 mL of acetone was added. The mixture was again placed in the centrifuge for 10 min at high speed. The supernatant solution was decanted and the precipitate was dried. Deionized water was added and mixed to completely dissolve the precipitate. The emission (excitation wavelength 405 nM) and absorbance (Spectrophotometer UV/VIS/NIR V-570 JASCO) spectra of the resulting solution were measured.

The presence of homogenous CDs was confirmed by transmission electron microscopy (TEM) picture (FEI Tecnai G2 X-TWIN, Dawson Creek Drive, Hillsboro, OR, USA). The Fourier Transform Infrared Spectroscopy (FT-IR) spectrum, using the Nicolet iS10 spectrometer (ThermoFisher, Waltham, MA, USA), was executed for observation of bonding between CDs/cysteamine (CA) and the enzyme.

In the present study, CDs are used as a semi-conducting material, which, thanks to the ability to improve the electron transport between the active site of the enzyme and the electrode surface, allows the sensor to present a short response time and high sensitivity. In addition, this material acts as a platform for protein immobilization. The solution containing synthesized CDs was stable for at least 6 months and stored at 4 °C when not used. The stability of CDs was confirmed by the emission measurements.

#### 2.2.2. Modification of Electrodes

The gold electrode (Au-E, Au electrode, diameter 2.5 mM, produced by BASi, MF-2014) was polished before the experiment with 3 µM fine diamond polish and rinsed thoroughly with double distilled water. The electrode was modified with a thin layer of CA, CDs and tyrosinase. CA solution (0.1 M) was applied onto the gold electrode for 24 h to link with the electrode surface by creating thiol bonds. CA molecule has an amine group, which stays exposed for the next step of modification. CDs have carboxylic groups with negative charge, which not only ensure good stability, but also enabled interaction with amine functional groups of CA and CDs. CA was physically adsorbed for 24 h to link with CA/CDs and thus created CDs-modified electrode (Figure 1). CDs, like most nanoparticles, possess semi-conducting properties which could make the surface of the electrode a more conductive and monodispersed surface as a platform for protein binding. The immobilization process was executed by physical adsorption of Tyr (2 mg/ml) in Phosphate Buffered Saline (PBS) buffer (0.1 M; pH 7.0) at room temperature onto the surface of Au electrode modified with CA/CDs for 2 h, and then crosslinked with 10% glutaraldehyde (GA) (10 min). Additional use of the GA cross-linking allows a relatively stable and active immobilization of the tyrosinase onto the CA/CDs surface, because of the creation of covalent bonds [29].

The excess of unbounded proteins was washed with phosphate (0.1 M; pH 7.0), acetate (0.1 M; pH 5.2) and Tris-HCl (0.25 M; pH 7.2) buffers.

An enzyme immobilized by physical adsorption with cross-linker does not require any other activation. Modified Au-E/CA/CDs/Tyr electrode was obtained and was stored at 4 °C when not used. The described modified electrode was active for c.a. 80 cycles.

#### 2.2.3. Electrochemical Measurements

All electrochemical experiments for NE determination were conducted using DPV and CV methods with a potentiostat/galvanostat AUTOLAB PGSTAT128N (serial nr. AUT84866; Utrecht, The Netherlands) with GPES software (version 4.9). A conventional three-electrode system was used for all electrochemical measurements in 8-mL cell. Gold electrode (unmodified or modified with CA/CDs/Tyr) was used as a working electrode, together with a coiled platinum wire as an auxiliary electrode and a silver-silver chloride reference electrode (Ag/AgCl). CV measurements were carried out by repeated potential scanning in range −0.2–1.2 V. DPV analysis was conducted with potential −0.2–0.6 V. All electrochemical measurements were performed at room temperature and in open-air conditions.

#### 2.2.4. Electrochemical Determination of Norepinephrine

NE detection was determined using DPV technique in 8-mL cell. NE tests solutions were prepared by dissolving NE in 0.1 M PBS buffer (pH 7.0). The measurements were conduct in 1–200 μM range and the current response was proportional to the proper concentration.

In addition, CV technique was employed for showing the whole enzyme-based redox reaction during NE determination. Such test was provided in potential range −0.2 V–1.2 V, with scan rate 50 mV/s, for 10 cycles each.

#### 2.2.5. Influence of Interfering Substances

Interfering species (ascorbic acid (AA), uric acid (UA), L-cysteine (CYS), dopamine (DA), epinephrine (EP) and 4-tert-Butylcatechol (4tBC)) were added to each NE standard solution in a concentration of 50 μM to investigate the selectivity of the proposed method. Listed compounds were mixed each time with NE solutions in the volume ratio 1:1.

## 3. Results and Discussion

### 3.1. Characterization of CDs

Figure 2A shows collected absorbance and photoluminescence spectra of CDs after synthesis. As can be seen, CDs have a maximum of emission at 503 nm, which corresponds to a green color. In order to confirm the presence of CDs, TEM measurements were performed (Figure 2B). The morphological display of the CDs shows that they were nearly homogeneous and monodisperse, with particle size of approximately 3 nm.

CDs are significant as a matrix for enzyme anchor. Synthesized CDs with carboxyl groups on their surface represent a suitable platform for amine groups present in the enzyme. Proteins can be successfully attached to CDs to initiate biorecognition element for NE detection.

Formation of specific functional groups in CDs (needed for anchoring the protein onto the surface of CDs), as well as formation of the bond between CDs and cysteamine and formation of the system (CA-CDs-Tyr) was confirmed by FT-IR spectroscopic analysis, as shown in Figure 3A,B. Because of the presence of aqueous media, samples were prepared on glass plates (background) using a layer-by-layer method.

The FT-IR spectrum of CA shows the bands at 3090 cm^−1^, which are attributed to N-H_2_ in the amine group. The same characteristic peaks were shifted to 2919 cm^−1^ on the CA-CDs spectrum, which shows the creation of amide group between CA and CDs [30]. Peaks from thiol group of CA (2516 cm^−1^) disappeared on the CA-CDs spectrum, which proves an interaction between the gold electrode and CA-CDs.

The peak at 3328 cm^−1^ on the CDs is characteristic for the hydroxyl group (−OH), and the peak at 1036 cm^−1^ corresponds to C-O interactions from the carboxylic group on the CDs surface. Additionally, peaks at 1569 and 1457 cm^−1^ come from an O-H interaction (Figure 3A) [31].

The disappearance of O-H peaks in the region near 1500 cm^−1^ on the CA-CDs-Tyr can suggest, that protein is immobilized on CDs-based platform successfully [32]. It can be assumed that there is a thorough electrode coverage. Although the enzyme influences the peak intensity, characteristic chemical bands of CA-CDs were preserved (Figure 3B).

### 3.2. Cyclic Voltammetric Behavior of NE

The electrochemical signals of NE at Au-E, Au-E/CA/CDs and Au-E/CA/CDs/Tyr were measured using CV in 200 μM NE solution in PBS buffer (pH 7.0). PBS buffer is the optimal buffer for tyrosinase activity. As shown in Figure 4A, at a bare Au electrode a pair of NE redox peaks occur, however low signals are obtained. After coating the Au electrode surface with CA and CDs, the redox peaks increased slightly. Nevertheless, for Au-E/CA/CDs/Tyr, a significant signal growth was observed, which was due to the unique properties (e.g., stability in time) of CA/CDs; the CA together with CDs and the catalyst, tyrosinase, showed the highest oxidation and reduction signals of NE. Therefore, such bio-platform was used for the determination of sensitivity towards NE. Negatively charged carboxyl groups make CDs a suitable matrix for protein immobilization. Application in the receptor part layer of a conductive nanomaterial allows improvement of the parameters of a biosensor, like a sensor sensitivity, increasing the limit of detection and lengthening the life of the sensor. The nanomaterial also provides a place for anchoring the protein while maintaining its catalytic activity. Tyrosinase belongs to electron transfer proteins; immobilization of Tyrosinase on the surface of CA/CDs can promote electron transfer between semi-conductive CA/CDs-film and the electrode, with simultaneous oxidation of NE. Additionally, to register the enzymatic activity of tyrosinase immobilized on modified electrode, supplementary measurements for detection system (Au-E/CA/CDs/Tyr) without NE in the investigated sample were conducted (Figure 4B).

### 3.3. Calibration and Limit of Detection of NE

Electrochemical signals of NE were investigated in a wide range of concentration (1–200 μM), employing the more sensitive DPV technique in oxygen-saturated conditions. Open-air conditions are in most cases crucial in the construction of such bio-tools, as these devices should work continuously in an oxygen-saturated state. According to Solomon et al. [13], the resting state is mainly in the oxidized form [Tyr–Cu(II)] which can interact with phenol derivatives, such as NE. The products of this catalytic reaction are the oxidized form of NE - *o*-quinone, and the reduced form of the enzyme [Tyr–Cu(I)], as shown in Figure 5A. In Figure 5B the produced *o*-quinone is present, which can be again changed into NE [33].

The main problem during phenol-derivatives (e.g., NE) electrochemical oxidation is the electrode surface deactivation. It is caused by the formation of a passivating-polymeric film produced by the coupling of electrogenerated phenoxy radical [34]. While providing electrochemical measurements (e.g., using CV technique), such deactivation may be visible as a decrease in the oxidation current and an increase in the oxidation potential, when consecutive cycles are performed on the same electrode [2]. In consequence, the sensor loses reproducibility of the measurements, which is one of the main parameters characterizing such tools. Due to this, to characterize the work of the biosensor, the linear range and detection limit were examined using DPV method. Figure 6A shows the oxidation current peaks (I_pa_) of NE increasing with its concentration (ranging from 1 μM to 200 μM). These signals precisely respond to the given NE concentration. Figure 6B presents a linear response to NE in the studied concentration range; good linear coefficient (R^2^ = 0.998) was observed. The biosensor represents an excellent linear response in a broad range of concentrations.

Limit of detection for described method (LOD) was calculated as (1) [35]:LOD = 3.29 σ_B_/b(1)
where σ_B_ is the standard deviation of the population of blank responses and b is the slope of the regression line.

LOD calculated this way was found at 196 nM. There are only a few reports in the literature describing nanomaterial-based biosensing electrochemical platforms for NE determination. However, this LOD value is comparable to or lower than published results [36,37,38,39] which illustrates that Au-E/CA/CDs/Tyr has good sensitivity in wide linear range (Table 1).

Limit of quantification (LOQ) was also determined (calculated using Equation 2) and it equals 312 nM.
LOQ = 5 σ_B_/b(2)
where σ_B_ is the standard deviation of the population of blank responses and b is the slope of the regression line [35].

Furthermore, sensitivity of proposed biosensor was found at 4.2 μA mM^−1^cm^−2^.

### 3.4. Selectivity

The selectivity is an extremely important parameter in designing bio-devices for diagnostic purposes. The tyrosinase-based biosensor presented here was developed for quantitative NE monitoring in human samples using DPV technique. In that case, the biosensor has to be selective only for NE detection, with minimal or no influence from other species present in the sample. Human body fluids contain a number of interfering substances, such as the most common ascorbic acid and uric acid. Figure 7 represents a wide range of species which may disturb NE signals, including ascorbic acid (AA), uric acid (UA), cysteine (CYS), epinephrine (EP), dopamine (DA) and 4-tert-Butylcatechol (4tBC). These compounds were added (50 μM) to every investigated NE sample (concentrations of NE 1, 50, 100 μM) to check an impact, while carrying out measurements into their high excess, equilibrium, and deficiency. L-DOPA, as a neurotransmitter and precursor of NE, has been selected as a possible interfering compound. What is more, its chemical structure is similar to NE, so it was necessary to check for possible effects on NE determination using the described procedure. The highest impact on the NE measurements had other neurotransmitters: EP and DA (21.25% and 18.5%, respectively), due to their similar structure to NE. It is known that DPV is an adequate technique for the analysis of the mixture of electrochemically active compounds, because a relatively small difference in their potential peak is needed. Despite the influence of EP and DA equal to c.a. 20%, the distinction of NE from other neurotransmitters and sensitive detection of NE was possible. Other species present negligible effects (<8.7%) on the current peak of the samples, compared to the blank (AA: 10.6%, UA: 5.55%, CYS: 4.59%, 4tBC: 14.15%). The presented results (Figure 7) confirm an insignificant impact on the selectivity of fabricated bio-tool, and prove that existing interferences do not interrupt the prominence of proposed NE test.

### 3.5. Real Application

The practicability of the introduced method was tested by the detection of NE dissolved in synthetic urine CLEANU^®^, containing, among others, creatinine, uric acid, mineral salts, dyes, urea and water (exact information reserved according to patent). Executed determination of the NE concentration in synthetic urine based on DPV method for three measurements showed an exquisite recovery value (Table 2). These results (calculated as ratio of detected concentration to the real concentration of norepinephrine in the synthetic urine (%)) demonstrate that the proposed strategy is selective, sensitive and suitable for real detection of NE.

## 4. Conclusion

In the present study, a new NE biosensing assay was developed and characterized. NE biosensor based on the Au-E/CA/CDs/Tyr bioplatform with high sensitivity, selectivity and practicability. The described method shows an exquisite electrocatalytic activity over a broad linear concentration range (1–200 × 10^−6^ M) with a detection limit of 196 nM, quantification limit of 312 nM and sensitivity of 4.2 μA mM^−1^cm^−2^. It is noteworthy that the modified electrode exhibited good selectivity when tested in a wide range of interfering compounds (AA, UA, CYS, 4tBC, EP, DA). In addition, the obtained biosensor successfully validates the proposed strategy, with adequate recovery, for NE detection in biological samples. All characteristics made for NE determination with Au-E/CA/CDs/Tyr establish a convenient, stable, simple and long-term technique and it can be recommended as an excellent analytical bio-device.

## Figures and Tables

**Figure 1 sensors-20-04567-f001:**
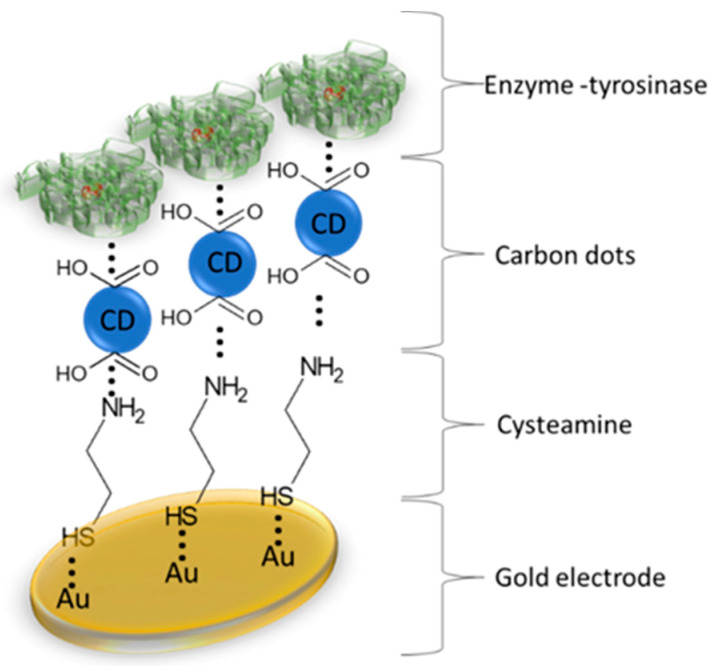
Scheme of measuring system of norepinephrine based on Gold-Electrode/Cysteamine/Carbon Dots/Tyrosinase.

**Figure 2 sensors-20-04567-f002:**
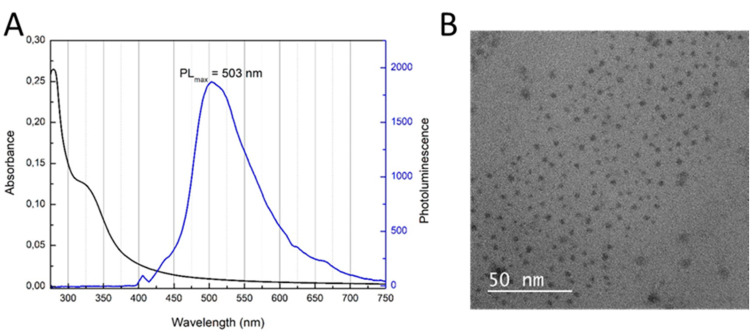
Characteristic information about CDs: (**A**): Absorbance and photoluminescence spectra, (**B**): Picture from TEM.

**Figure 3 sensors-20-04567-f003:**
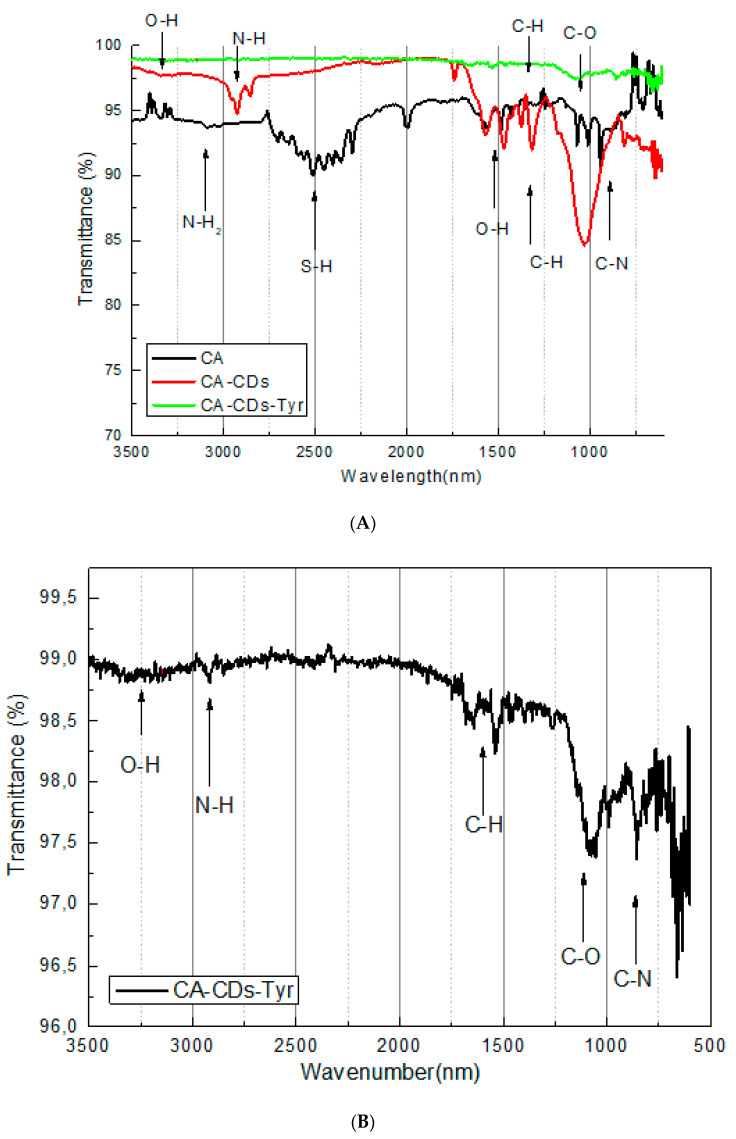
FT-IR spectrum of (**A**): cysteamine (CA), Carbon dots attached to cysteamine (CA-CDs), system with enzyme (CA-CDs-Tyr) and (**B**): enlargement of the system.

**Figure 4 sensors-20-04567-f004:**
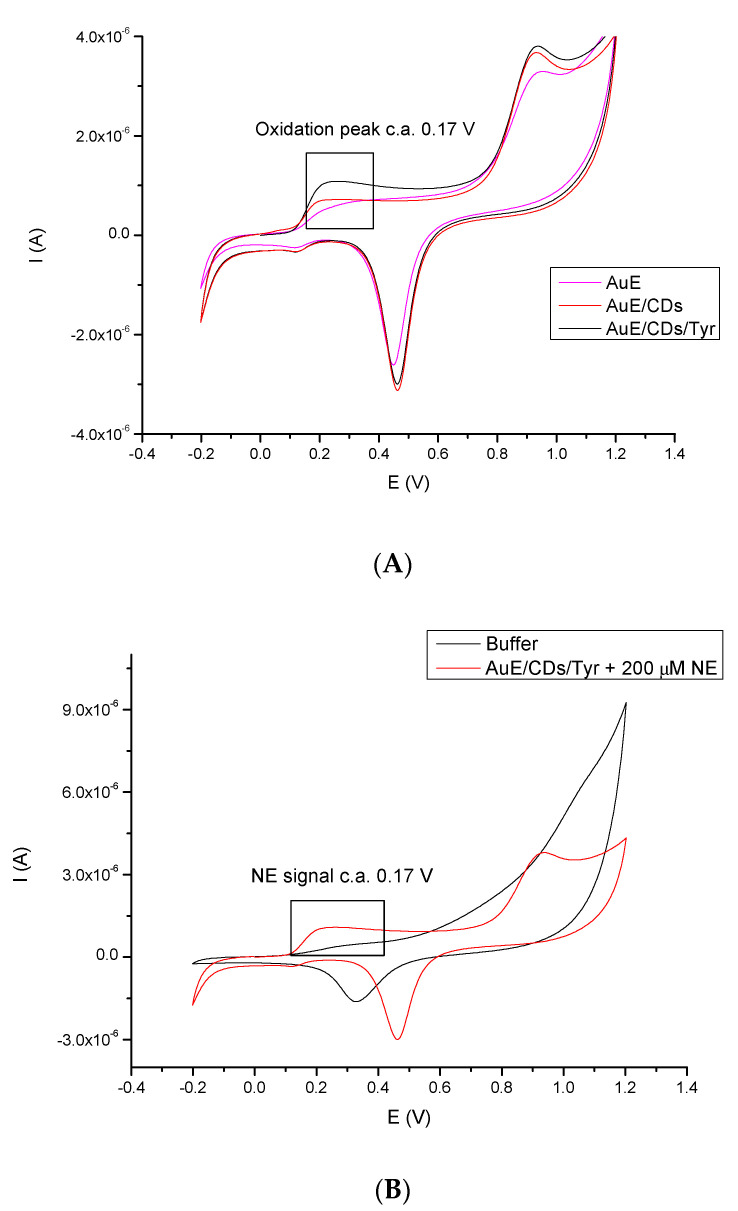
(**A**): Representative CV-scans of the bare Au electrode, Au electrode modified with cysteamine and carbon dots, and Au-E/CA/CDs/Tyr system in the presence of norepinephrine (200 μM) under applied potential in range −0.2–1.2 V, scan rate 50 mV/s vs. Ag/AgCl (0.1 M); (**B**): Representative CVs of the Au-E/CA/CDs/Tyr in the absence and in the presence of 200 μM NE in buffer solution (pH 7.0, scan rate 50 mV/s, applied potential: −0.2–1.2 V vs. Ag/AgCl 0.1 M).

**Figure 5 sensors-20-04567-f005:**
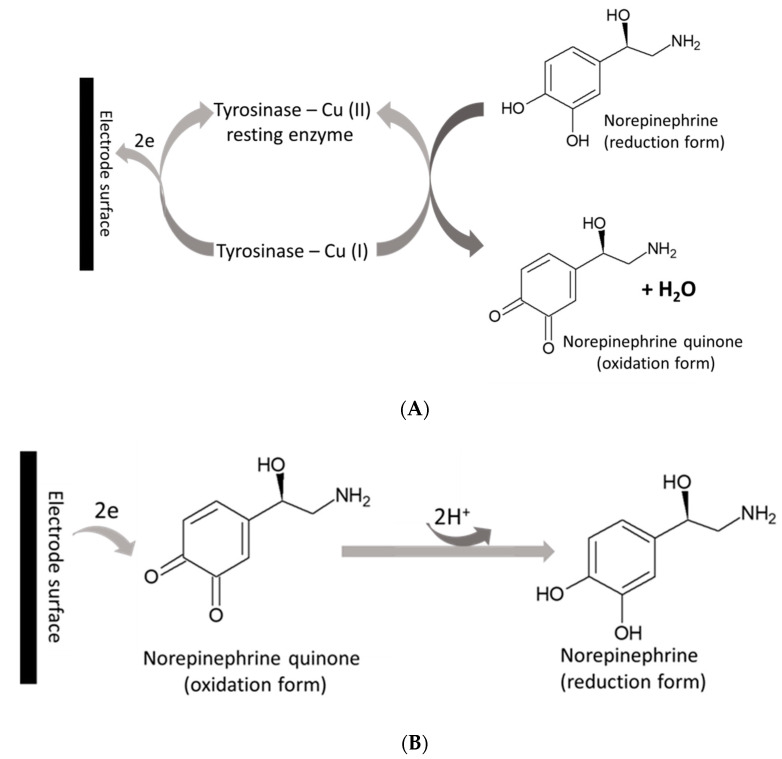
(**A**,**B**). Schematic representation of electron transfer between Tyr and Au-modified electrode (**A**): anodic reaction and (**B**): cathodic reaction.

**Figure 6 sensors-20-04567-f006:**
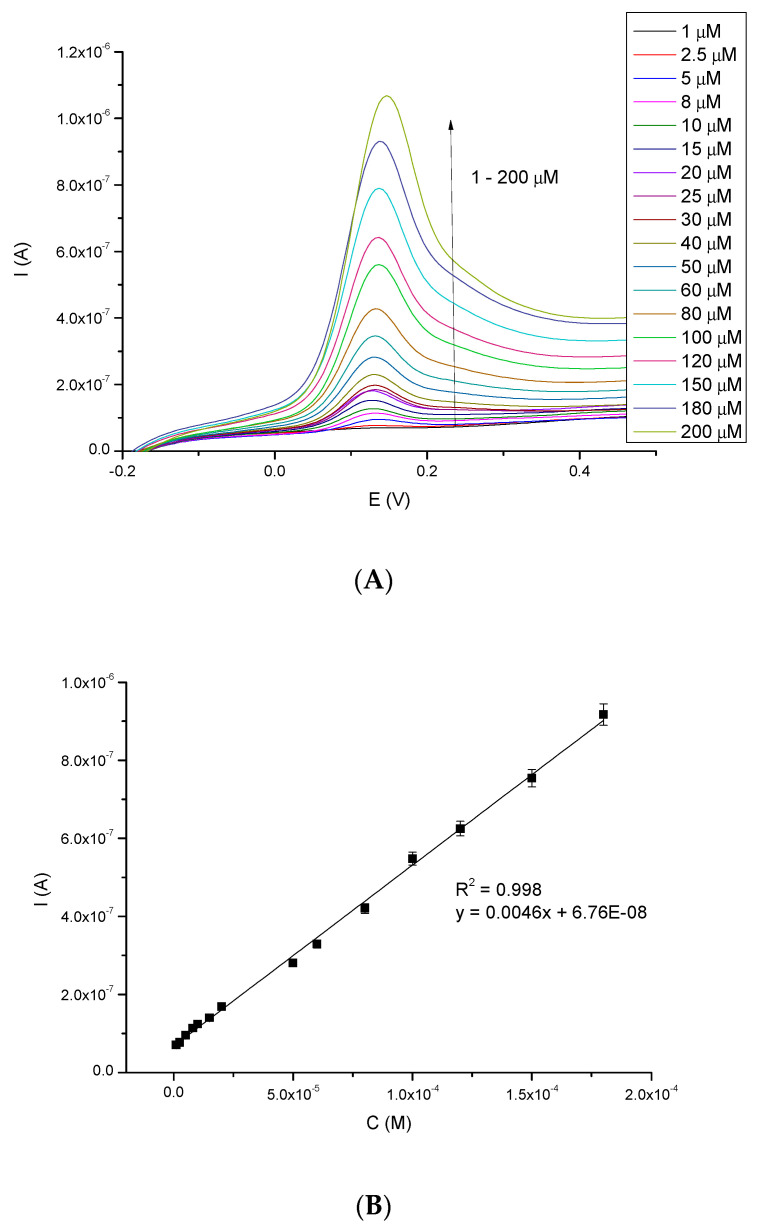
(**A**): Differential Pulse Voltammetry-scans for different concentrations of NE in range 1–200 μM vs. Ag/AgCl (0.1 M) electrode and (**B**): Linear relationship between current and NE concentration (1–200 μM).

**Figure 7 sensors-20-04567-f007:**
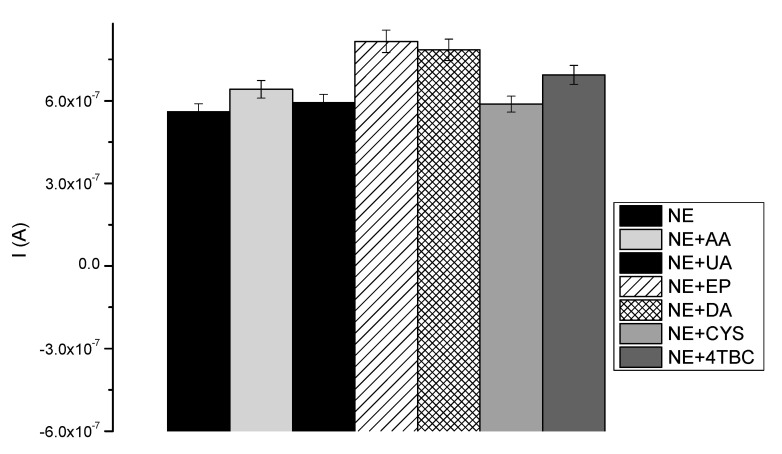
Effect of interfering substances (50 μM) on NE determination.

**Table 1 sensors-20-04567-t001:** Comparison of biosensors and sensors for NE determination.

	Biosensor/Sensor	Technique	Linear Range	LOD	Ref.
1	GCE/ECR *	CV, DPV	2–50 μM	1.5 μM	[36]
2	CPE/BH/TiO_2_ **	DPV	4–1100 μM	0.5 μM	[37]
3	GCE/DNA/AuNPs ***	DPV	0.5–80 μM	5 nM	[38]
4	CPE/NMM ****	CV, DPV	0.07–2000 μM	0.04 μM	[39]
5	Au-E/Cys/CDs/Tyr	DPV	1–200 μM	196 nM	This work

* GCE—glassy carbon electrode, ECR—Eriochrome Cyanine R. ** CPE—carbon paste electrode, BH—2,2′-[1,2 buthanediyl bis(nitriloethylidyne)]-bis-hydroquinone. *** AuNPs—gold nanoparticles. **** NMM—nanostructured mesoporous material.

**Table 2 sensors-20-04567-t002:** Results obtained for NE determination based on proposed method.

Concentration of NE in Real Sample [μM]	C_detected_ [μM]	Recovery [%] (Average)	RSD (Calculated for 20 Repetitions)
100.00	98.44	98.70	±0.73
100.00	99.11		
100.00	98.57		

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
