# Peer review of "Simple and Cost-Effective Electrochemical Method for Norepinephrine Determination Based on Carbon Dots and Tyrosinase"

_sensors, 2020, doi:10.3390/s20164567_

Round 1
Reviewer 1 Report
On the one hand, this is an interesting report, which is well organized. Moreover, the experimental results seems to be solid. On the other hand, there are serious problems in the work under review. First and foremost, the authors do not fully understand redox reactions occurring during the function of the developed biosensor. Thus, despite the fact that experimental part of the manuscript is solid, I do not recommend publication if the below mentioned problems are not solved.
Lines 32-34: ”According to Cammann, biosensors are analytical devices allowing for conversion of biological signal to an electrical signal, e.g. amperometric response in case of electrochemical sensors [1]”
This is just partially true. There are potentiometric biosensors, as well as biodevices based on electrochemical impedance spectroscopy. Please, modified the sentence accordingly and provide additional references, if needed.
Lines 60-66: “The main problem while designing enzymatic biosensor is to increase the speed and reversibility of charge transfer between the enzyme and the electrode. A direct electron transfer can be optimized using conductive materials as templates for immobilization of proteins. Biosensors built on semiconductor nano- and microstructures are one of the most attractive science applications in the last decade. This kind of materials, e.g. nanoparticles such as carbon dots, improve the electron transport between the active site of the enzyme and the electrode surface, which increases the sensitivity of the sensor and shortens the response time [15].”
This paragraph should be either deleted or completely re-written. First, in the manuscript under review the authors presented a second generation biosensor, which has nothing to do with direct electron transfer, charge transfer between the enzyme and the electrode, ET between the active site of the enzyme and the electrode surface, etc. Second, direct electron transfer based bioelectrocatalytic reactions on tyrosinase modified electrodes are questionable. Additional details can be found in Refs. Shleev et al. Biosensor and Bioelectronics, 2005,20, 2517.
Line 99: “Tyrosinase (from mushroom, …”
I assume from fungus? Which one? This should be clarified.
Lines 247-249: ” PBS buffer is composed of the substances close to human serum, therefore creating a good environment in case of real sample study.”
This is not true. PBS just has similar pH and ionic strength. Human serum is a complex fluid. Thus, the sentence should be either deleted or re-written.
Lines 245-274: ”3.2. Cyclic voltammetric behavior of NE”.
This is one of the most important and tricky parts of the manuscript. As far as I understand the mechanism of the developed biosensor, the substrate (bioanalyte, i.e. NE) is oxidized by tyrosinase and NE-quinone is reduced at the electrode surface back to NE. This is a typical mechanism for second generation enzymatic sensors. However, in Item 3.2. the authors are focused on the oxidation peak, whereas from electrochemical point of view the reduction peak is the most important part of CVs. Again, oxidation of NE should be enzymatic, not electrochemical. Moreover, the authors wrote “Tyrosinase is a common electron transfer protein, and such protein immobilized on the surface of CA/CDs can promote electron transfer between semi-conductive CA/CDs-film and the electrode.” First, I do not understand this sentence. Second, tyrosinase is not an electron transfer protein, as azurin, cyt c, etc. It is a redox enzyme catalyzing reactions of substrate oxidation by molecular oxygen. To make a long story short, it seems like the authors do not fully understand redox reactions occurring during the function of the biodevice, as also reflected by the presence of unnecessary paragraph in Introduction (vide supra) and questionable scheme presented in Fig. 5 (vide infra).
Lines 287-291: ”Fig. 5. Mechanism of norepinephrine redox reaction catalyzed by tyrosinase”.
Fig. 5 should be re-drawn. First, enzymatic oxidation of norepinephrine redox reaction catalyzed by tyrosinase is irreversible process. Second, the reduction of norepinephrine quinone occurs at the electrode surface, which is not visualized on the current scheme. Also, I recommend combining Fig. 5 with Fig. 1.
General comments:
- Please, avoid usage “very”, “often”, “a little”, etc. These terms are not scientific.
- One could cite and discuss similarities and differences between your work and one of recently published papers regarding electrochemical norepinephrine detection, viz. “Smartphone-based square wave voltammetry system with screen-printed graphene electrodes for norepinephrine detection” by Daizong Ji, Zhouyuanjing Shi, et al. Smart Materials in Medicine, 2020, 1, 1-9.
Author Response
Response to Reviewer 1 Comments
We would like to thank for Reviewer’1 detailed review, in addition we would like to inform, that every significant changes in the manuscript have been highlighted (green colour). Also, English language and grammar mistakes have been checked and improved.
Point 1: Lines 32-34: ”According to Cammann, biosensors are analytical devices allowing for conversion of biological signal to an electrical signal, e.g. amperometric response in case of electrochemical sensors [1]”
This is just partially true. There are potentiometric biosensors, as well as biodevices based on electrochemical impedance spectroscopy. Please, modified the sentence accordingly and provide additional references, if needed.
Response 1: Thank you for this comment, of course you are right and we changed this sentence as: ”According to Cammann, biosensors are analytical devices allowing for conversion of biological signal to a measurable signal, like for instance amperometric response in case of electrochemical sensors [1]”
Point 2: Lines 60-66: “The main problem while designing enzymatic biosensor is to increase the speed and reversibility of charge transfer between the enzyme and the electrode. A direct electron transfer can be optimized using conductive materials as templates for immobilization of proteins. Biosensors built on semiconductor nano- and microstructures are one of the most attractive science applications in the last decade. This kind of materials, e.g. nanoparticles such as carbon dots, improve the electron transport between the active site of the enzyme and the electrode surface, which increases the sensitivity of the sensor and shortens the response time [15].”
This paragraph should be either deleted or completely re-written. First, in the manuscript under review the authors presented a second generation biosensor, which has nothing to do with direct electron transfer, charge transfer between the enzyme and the electrode, ET between the active site of the enzyme and the electrode surface, etc. Second, direct electron transfer based bioelectrocatalytic reactions on tyrosinase modified electrodes are questionable. Additional details can be found in Refs. Shleev et al. Biosensor and Bioelectronics, 2005, 20, 2517.
Response 2: Mentioned paragraph has been deleted.
Point 3: Line 99: “Tyrosinase (from mushroom, …”
I assume from fungus? Which one? This should be clarified.
Response 3: Tyrosinase from Agaricus bisporus. We are truly sorry for this mistake.
Point 4: Lines 247-249: ” PBS buffer is composed of the substances close to human serum, therefore creating a good environment in case of real sample study.”
This is not true. PBS just has similar pH and ionic strength. Human serum is a complex fluid. Thus, the sentence should be either deleted or re-written.
Response 4: This sentence has been deleted.
Point 5: Lines 245-274: ”3.2. Cyclic voltammetric behavior of NE”.
This is one of the most important and tricky parts of the manuscript. As far as I understand the mechanism of the developed biosensor, the substrate (bioanalyte, i.e. NE) is oxidized by tyrosinase and NE-quinone is reduced at the electrode surface back to NE. This is a typical mechanism for second generation enzymatic sensors. However, in Item 3.2. the authors are focused on the oxidation peak, whereas from electrochemical point of view the reduction peak is the most important part of CVs. Again, oxidation of NE should be enzymatic, not electrochemical. Moreover, the authors wrote “Tyrosinase is a common electron transfer protein, and such protein immobilized on the surface of CA/CDs can promote electron transfer between semi-conductive CA/CDs-film and the electrode.” First, I do not understand this sentence. Second, tyrosinase is not an electron transfer protein, as azurin, cyt c, etc. It is a redox enzyme catalyzing reactions of substrate oxidation by molecular oxygen. To make a long story short, it seems like the authors do not fully understand redox reactions occurring during the function of the biodevice, as also reflected by the presence of unnecessary paragraph in Introduction (vide supra) and questionable scheme presented in Fig. 5 (vide infra).
Response 5: According to [B. C. Janegitz, R. A. Medeiros, R. C. Rocha-Filho, O. Fatibello-Filho, Direct electrochemistry of tyrosinase and biosensing for phenol based on gold nanoparticles electrodeposited on a boron-doped diamond electrode. Diamond and Related Materials 25 (2012) 128–133; Y. He, X. Yang, Q. Han, J. Zheng, The Investigation of Electrochemistry Behaviors of Tyrosinase Based on Directly-Electrodeposited Graphene on Choline-Gold Nanoparticles. Molecules, 22 (2017) 1-12] tyrosinase is a direct electron transfer protein. It catalyzed redox reaction of phenol and amine derivatives (as NE) by electron transfer with molecular oxygen. A few papers about the direct electrochemistry of tyrosinase were reported in the literature [A. Mohammadi, A.B. Moghaddam, R. Dinarvand, S. Rezaei-Zarchi, Direct electron transfer of polyphenol oxidase on carbon nanotube surfaces: application in biosensing, International Journal of Electrochemistry Science 4 (2009) 895–905; D.M. Sun, C.X. Cai, W. Xing, T.H. Lu, Immobilization and direct electrochemistry of copper-containing enzymes on active carbon, Chinese Science Bulletin 49 (23) (2004) 2452–2454; A.I. Yaropolov, A.N. Kharybin, J. Emneus, G. Marko Varga, L. Gorton, Electrochemical properties of some copper-containing oxidases, Bioelectrochemistry and Bioenergetics 40 (1996) 49–57; B.X. Ye, X.Y. Zhou, Direct electrochemical redox of tyrosinase at silver electrodes, Talanta 44 (1997) 831–836; H. Zhou, L. Liu, K. Yin, S.L. Liu, G.X. Li, Electrochemical investigation on the catalytic ability of tyrosinase with the effect of nano titanium dioxide, Electrochemistry Communications 8 (2006) 1168–1172], however they suffer from some defects. Unfortunately, redox peaks, corresponding to the T3 site, could not be seen on cyclic voltammetry (CV) scans performed under anaerobic conditions [Reuillard, B.; Goff, A.L.; Agnès, C.; Zebda, A.; Holzinger, M.; Cosnier, S. Direct electron transfer between tyrosinase and multi-walled carbon nanotubes for bioelectrocatalytic oxygen reduction. Electrochem. Commun. 2012, 20, 19–2]. Thus, the direct electrochemistry of Tyr is commonly difficult to be observed. There are only a few studies that described the direct electrochemistry of Tyr based on Woodward’s Reagent K [Faridnouri, H.; Ghourchian, H.; Hashemnia, S. Direct electron transfer enhancement of covalently bound tyrosinase to glassy carbon via Woodward’s reagent K.Bioelectrochemistry2011,82, 1–9] or single-walled carbon nanotubes [Mohammadi, A.; Moghaddam, A.B.; Dinarvand, R.; Rezaei-Zarchi, S. Direct electron transfer of polyphenoloxidase on carbon nanotube surfaces: Application in biosensing. Int. J. Electrochem. Sci. 2009, 4, 895–905]. NE at low concentrations can be detected via the increase of current that occurs associated to DET; in this sense, adequate biosensor construction is very important to obtain a good response between the electrode surface and the enzyme [B. C. Janegitz, R. A. Medeiros, R. C. Rocha-Filho, O. Fatibello-Filho, Direct electrochemistry of tyrosinase and biosensing for phenol based on gold nanoparticles electrodeposited on a boron-doped diamond electrode. Diamond and Related Materials 25 (2012) 128–133]. Due to this it seems to necessary use the support matrix with desire properties, as improvement of electron transfer, such as carbon dots [Campuzano, S., Yáñez-Sedeño, P., & Pingarrón, J. M. Carbon Dots and Graphene Quantum Dots in Electrochemical Biosensing. Nanomaterials, 9 (2019) 1-18]. Carbon can act as a primary electron donor to native electroactive sites of enzymes [El Kaoutit, M.; Naranjo-Rodriguez, I.; Temsamani, K.R.; Domínguez, M.; Hidalgo-Hidalgo de Cisneros, J.L. A comparison of three amperometric phenoloxidase-Sonogel-Carbon based biosensor for determination of polyphenols in beers. Talanta 2008, 75, 1348–1355]. In DPV analysis the oxidation peak is taking into consideration, this is why, the authors showed in CV voltammogram (Figure 4) the oxidation peak of NE (which signal is growing with using CDs as well as CDs and tyrosinase). CV presents only the whole redox reaction process, however all detection was conducted with DPV method. It is an enzymatic redox reaction, which was visualized with CV technique.
Point 6: Lines 287-291: ”Fig. 5. Mechanism of norepinephrine redox reaction catalyzed by tyrosinase”.
Fig. 5 should be re-drawn. First, enzymatic oxidation of norepinephrine redox reaction catalyzed by tyrosinase is irreversible process. Second, the reduction of norepinephrine quinone occurs at the electrode surface, which is not visualized on the current scheme. Also, I recommend combining Fig. 5 with Fig. 1.
Response 6: Figure 5 has been changed in manuscript. According to Solomon et al. [E.I. Solomon, U.M. Sundaram, T.E. Machonkin, Multicopper oxidases andoxygenases, Chem. Rev. 96 (1996) 2563–2605], the resting state is mainly in the oxidized form [Tyr–Cu(II)] which can interact with phenol derivatives, as NE. The products of this catalytic reaction is the oxidized form of NE - o-quinone, and the reduced form of the enzyme [Tyr–Cu(I)]. In scheme 5B is present the produced o-quinone, which can be changed into NE [Faridnouri, H., Ghourchian, H., & Hashemnia, S. (2011). Direct electron transfer enhancement of covalently bound tyrosinase to glassy carbon via Woodward’s reagent K. Bioelectrochemistry, 82(1), 1–9. doi:10.1016/j.bioelechem.2011.04.004].
|
|
Fig. 5 A,B. Schematic representation of electron transfer between Tyr and Au-modified electrode A: anodic reaction and B: cathodic reaction
Point 7: General comments:
- Please, avoid usage “very”, “often”, “a little”, etc. These terms are not scientific.
- One could cite and discuss similarities and differences between your work and one of recently published papers regarding electrochemical norepinephrine detection, viz. “Smartphone-based square wave voltammetry system with screen-printed graphene electrodes for norepinephrine detection” by Daizong Ji, Zhouyuanjing Shi, et al. Smart Materials in Medicine, 2020, 1, 1-9.
Response 7: We are truly sorry for this unscientific wording. Manuscript has been improved. Discussion is present in the manuscript.
At most recent work, D. Ji et. al. demonstrated an electrochemical, smartphone-based system for NE detection [Ji, D., Shi, Z., Liu, Z., Low, S. S., Zhu, J., Zhang, T., … Liu, Q. (2020). Smartphone-based square wave voltammetry system with screen-printed graphene electrodes for norepinephrine detection. Smart Materials in Medicine, 1, 1–9]. Screen-printed graphene electrodes were modified, e.g. with GOx, and used as a working electrodes. Described system did not use any biologically active compound. NE has been determined with square wave voltammetry (SWV) in concentration range 1 – 30 µM. The detection limit obtained for such system was equal to 265 nM. Selectivity study has been also provided. In comparison with our enzyme-based system, described sensor has slightly higher LOD and narrower linear range. In addition, due to enzyme application in recognition part of working electrode, biosensor present high selectivity. The main advantage of described system is facility of miniaturization and possibility of constant measurement, as a possible wearable device. It also presents the ability to work over a longer period of time (in comparison with biological systems).
Reviewer 2 Report
The authors present in this manuscript a work related to the electrochemical determination of norepinephrine using carbon dots and tyrosinase. Although, the developed sensor exhibit very good analytical parameters in terms of sensitivity, detection limit, and selectivity, this work should not be published in its present form unless a deep discussion on obtained results is reported. Indeed, the modification of the electrode with carbon dot and tyrosinase seems to be not necessary according to the results of figure 4. Modified electrode with Carbon dot showed a small oxidation peak at +0,17V (Figure 4 A) and thus experiments of DPV of AuE/CDs should be done and the obtained results should be compared with those obtained in presence of enzyme as depicted in figure 6. The authors claim that CDs facilitate the electron transfer as results of its specific surface area and many edge sites, however, the peak separation (the difference between oxidation and reduction peaks) of norepinephrine was unvaried as indicated in figure 4A.
Concerning selectivity, L-DOPA is supposed to interfere in the norepinephrine measurement. The authors should discuss this point.
Error bars should be indicated in all figures. The number of repetitions used in order to calculate the RSD of Table 1 should be reported.
A comparison with works already published in the literature in connection with norepinephrine sensor should be discussed.
Author Response
Response to Reviewer 2 Comments
We would like to thank for Reviewer’s 2 detailed review. In addition, we inform, that every significant changes in the manuscript have been highlighted (violet colour).
Point 1: Indeed, the modification of the electrode with carbon dot and tyrosinase seems to be not necessary according to the results of figure 4. Modified electrode with Carbon dot showed a small oxidation peak at +0,17V (Figure 4 A) and thus experiments of DPV of AuE/CDs should be done and the obtained results should be compared with those obtained in presence of enzyme as depicted in figure 6. The authors claim that CDs facilitate the electron transfer as results of its specific surface area and many edge sites, however, the peak separation (the difference between oxidation and reduction peaks) of norepinephrine was unvaried as indicated in figure 4A.
Response 1: The organization of an enzyme-based biosensor requires the integration of the biocatalyst with the support or immobilized materials to the extent that the biocatalytic transformation is, for instance, electronically transduced. What is more, immobilization can help to extend the lifetime of enzymes [Choi, M. M. F. (2004). Progress in Enzyme-Based Biosensors Using Optical Transducers. Microchimica Acta, 148(3-4), 107–132]. According to reconstituted oxidoreductase with nanoparticles showed higher activities than native enzymes with the natural electron acceptor, oxygen. The unusually higher enzyme activity was attributed to the enhanced efficiency of electron conduction via the concrete nanoparticles (Au, carbon, graphite). CDs are used for the design of electrochemical biosensors, with special attention to representative electrochemical affinity biosensing platforms that take advantage of the excellent properties offered by these nanomaterials either as electrode modifiers or signal tags. The most remarkable advantages offered by these carbon nanomaterials in the development of electrochemical (bio)sensors compared to the widely used CNTs arise from their many surface functional groups, such as carboxyl, which impart better solubility in many solvents and aqueous media and facilitate their easy functionalization with organic, polymeric, inorganic, or biological species without further modification and/or activation [Campuzano, S., Yáñez-Sedeño, P., Pingarrón, J. M. (2019). Carbon Dots and Graphene Quantum Dots in Electrochemical Biosensing. Nanomaterials, 9(4), 634]. The immobilization of protein onto suitable platform is a crucial part during biosensing tool manufacturing. Authors decided to use nanomaterial based on carbon, due to the large number of advantages [Atkin, P.; Daeneke, T.; Wang, Y.; Careya, B.J.; Berean, K.J.; Clark, R.M.; Ou, J.Z.; Trinchi, A.; Cole, I.S.; Kalantar-Zadeh, K. 2D WS2/Carbon Dot Hybrids with Enhanced Photocatalytic Activity. J. Mater. Chem. A 2016, 4, 13563–13571; Martínez-Periñán, E.; Bravo, I.; Rowley-Neale, S.J.; Lorenzo, E.; Banks, C.E. Carbon nanodots as electrocatalysts towards the oxygen reduction reaction. Electroanalysis 2018, 30, 436–444; Li, H.; Chen, L.; Wu, H.; He, H.; Jin, Y. Ionic liquid-functionalized fluorescent carbon nanodots and their applications in electrocatalysis, biosensing, and cell imaging. Langmuir 2014, 30, 15016–15021; Ji, H.; Zhou, F.; Gu, J.; Shu, C.; Xi, K.; Jia, X. Nitrogen-doped carbon dots as a new substrate for sensitive glucose determination. Sensors 2016, 16, 630; Wang, Y.; Wang, Z.; Rui, Y.; Li, M. Horseradish peroxidase immobilization on carbon nanodots/CoFe layered double hydroxides: Direct electrochemistry and hydrogen peroxide sensing. Biosens. Bioelectron. 2015, 64, 57–62; Huang, Q.; Hu, S.; Zhang, H.; Chen, J.; He, Y.; Li, F.; Weng, W.; Ni, J.; Bao, X.; Lin, Y. Carbon dots and chitosan composite film based biosensor for the sensitive and selective determination of dopamine. Analyst 2013, 138, 5417–5423]. What is more, they act as an electron acceptors or donors to native electroactive sites of enzymes, improving the electrochemical determination of NE and are very efficient for the immobilization of increased loadings of proteins, enzymes (exhibiting DET), like tyrosinase [El Kaoutit, M.; Naranjo-Rodriguez, I.; Temsamani, K.R.; Domínguez, M.; Hidalgo-Hidalgo de Cisneros, J.L. A comparison of three amperometric phenoloxidase-Sonogel-Carbon based biosensor for determination of polyphenols in beers. Talanta 2008, 75, 1348–1355].
Obtained oxidation and reduction peaks (in potential 0.17 V and 0.1 V, respectively) increase (value of current is higher). Thanks to that, smaller concentration of NE can be detected. Due to this, carbon dots, improve the operational parameters of the sensor, however, they also act as a matrix for an enzyme immobilization. Prepared in described way CDs – matrix - are cheap, easy to produce, and stable.
Point 2: Concerning selectivity, L-DOPA is supposed to interfere in the norepinephrine measurement. The authors should discuss this point.
Response 2: L-DOPA, as a neurotransmitter and precursor of NE, has been selected as a possible interfere compound. What is more, its chemical structure is similar to NE, so authors wanted to check possible effect to NE determination using described procedure.
Point 3: Error bars should be indicated in all figures. The number of repetitions used in order to calculate the RSD of Table 1 should be reported.
Response 3: All error bars has been added to the figures, where needed, as well as, the number of repetitions needed for RSD calculation (20).
Point 4: A comparison with works already published in the literature in connection with norepinephrine sensor should be discussed.
Response 4: A comparison with other, similar works has been add in manuscript (in the table form).
|
Biosensor/Sensor |
Technique |
Linear Range |
LOD |
Ref. |
1 |
GCE/ECR* |
CV, DPV |
2–50 μM |
1.5 μM |
[36] |
2 |
CPE/BH/TiO2** |
DPV |
4.0–1100.0 μM |
0.5 μM |
[37] |
3 |
GCE/DNA/AuNPs*** |
DPV |
5×10−7 - 8×10−5M |
5 nM |
[38] |
4 |
CPE/NMM**** |
CV, DPV |
7.0 × 10−8 - 2.0 × 10−3 M |
4.0 × 10−8M |
[39] |
5 |
Au-E/Cys/CDs/Tyr |
CV, DPV |
1 - 200 μM |
196 nM |
This article |
* GCE – glassy carbon electrode, ECR - Eriochrome Cyanine R
**CPE-carbon paste electrode, BH- 2,2′-[1,2 buthanediyl bis(nitriloethylidyne)]-bis-hydroquinone
*** AuNPs - gold nanoparticles
**** NMM -nanostructured mesoporous material
[36] Yao H., Li S., Tang Y., Chen Y., Chen Y., Lin, X. ). Selective oxidation of serotonin and norepinephrine over eriochrome cyanine R film modified glassy carbon electrode, Electrochimica acta, 2009, 54(20), 4607- 4612.
[37] Mazloum-Ardakani M., Beitollahi H., Sheikh-Mohseni M. A., Naeimi H., Taghavinia, N., Novel nanostructure electrochemical sensor for electrocatalytic determination of norepinephrine in the presence of high concentrations of acetaminophene and folic acid, Applied Catalysis A: General, 2010, 378(2), 195-201.
[38] Lu L. P., Wang S. Q., Lin, X. Q., Fabrication of layer-by-layer deposited multilayer films containing DNA and gold nanoparticle for norepinephrine biosensor, Analytica chimica acta, 2004, 519(2), 161-166.
[39] Mazloum-Ardakani M., Sheikh-Mohseni M. A., Abdollahi-Alibeik M., Benvidi, A., Electrochemical sensor for simultaneous determination of norepinephrine, paracetamol and folic acid by a nanostructured mesoporous material. Sensors and Actuators B: Chemical, 2012, 171, 380-386.
Reviewer 3 Report
Baluta et al. presents a DPV based electrochemical method for the detection of norepinephrine. The electrochemical transducer layer was composed of cysteamine, carbon dots and tyrosinase. Practicability was validated by the detection of NE dissolved in synthetic urine. Overall, this manuscript fits well to the scope of Sensors. I have some minor issues for authors to address which are as follows:
1. In Materials and Methods 2.2.2, the authors introduced much information about the principles of the functionalization. However, important experimental details, including the concentrations of CA, GA, Tyr, and pH of the used solutions, are missing.
2. As stated by the authors, the developed biosensor showed a linear range between 1 - 200 μM, and a limit of detection at 196 nM. How would the authors envision its application in real biological samples where the norepinephrine levels are at nM level?
Author Response
We would like to thank for Reviewer’s 3 detailed review, in addition we would like to inform, that every significant changes in the manuscript have been highlighted (yellow colour).
Point 1: In Materials and Methods 2.2.2, the authors introduced much information about the principles of the functionalization. However, important experimental details, including the concentrations of CA, GA, Tyr, and pH of the used solutions, are missing.
Response 1: Thank you for this comment – concentrations and pH values should be present in manuscript. Here, we are presented missing values: concentrations of CA: 0.1 M; GA: 10 %; Tyr: 2 mg/ml and values of buffers: PBS buffer: 0.1 M, pH 7.0; acetate: 0.1 M; pH 5.4; phosphate: 0.1 M; pH 7.0, Tris-HCl: 0.25 M; pH 7.2. All missing informations have been added in the manuscript.
Point 2: As stated by the authors, the developed biosensor showed a linear range between 1 - 200 μM, and a limit of detection at 196 nM. How would the authors envision its application in real biological samples where the norepinephrine levels are at nM level?
Response 2: Due to [Wilke, N., Janßen, H., Fahrenhorst, C., Hecker, H., Manns, M. P., Brabant, E.-G., Breitmeier, D. (2007). Postmortem determination of concentrations of stress hormones in various body fluids—is there a dependency between adrenaline/noradrenaline quotient, cause of death and agony time? International Journal of Legal Medicine, 121(5), 385–394; E. Wierzbicka, M. Szultka-Młynska, B. Buszewski, G. D. Sulka, Sensors and Actuators B, vol. 237, 2016, 206–215] norepinephrine concentration in body fluids is c.a. 0.08 nM, however, for instance, in cardiological disorders this level may reach even 500 nM or more. Obtained by us LOD for NE detection is suitable for real application in body fluids analysis – prepared by us test can signalize that some disorders are present or starts in patient body. In addition, such device in presented linear range could serve as a tool for permanent monitoring of progress in treatment with this neurotransmitter.

Round 2
Reviewer 2 Report
The revised version of the manuscript is much improved. The authors should discuss in the text of the manuscript, the interference of L-DOPA. Experiments are not mandatory but the authors should not ignore in their study to talk about L-DOPA, as a neurotransmitter and precursor of NE.
Author Response
Dear Reviewer,
Thank you very much. Of course we improved our manuscript with this information.
Kind regards,
Joanna Cabaj